# Mucosal Relapse of Visceral Leishmaniasis in a Child with SARS-CoV-2 Infection

**DOI:** 10.3390/pathogens12091127

**Published:** 2023-09-03

**Authors:** Claudia Colomba, Giovanni Boncori, Chiara Albano, Valeria Garbo, Sara Bagarello, Anna Condemi, Salvatore Giordano, Antonio Cascio

**Affiliations:** 1Department of Health Promotion, Maternal and Infant Care, Internal Medicine and Medical Specialties, University of Palermo, 90100 Palermo, Italy; claudia.colomba@libero.it (C.C.); boncori.giovanni@yahoo.it (G.B.); vali.garbo@gmail.com (V.G.); sbagarello@gmail.com (S.B.); annacondemi96@gmail.com (A.C.); antonio.cascio03@unipa.it (A.C.); 2Division of Pediatric Infectious Diseases, “G. Di Cristina” Hospital, ARNAS Civico Di Cristina Benfratelli, 90100 Palermo, Italy; giordano.s@tiscali.it; 3Infectious and Tropical Diseases Unit, AOU Policlinico “P. Giaccone”, 90100 Palermo, Italy

**Keywords:** visceral, leishmaniasis, mucosal, relapse, pediatric

## Abstract

Leishmaniasis is a vector-borne disease caused by protozoan parasites of the genus Leishmania and is transmitted through the bite of infected female sandflies. In the Mediterranean region, visceral leishmaniasis is caused by *Leishmania. infantum,* and it is usually responsible for symptoms such as fever, pancytopenia and enlargement of the liver and spleen. Relapse is rare in immunocompetent patients as much as the mucous involvement. We present a rare case of mucosal relapse of visceral leishmaniasis in a child with SARS-CoV-2 infection and perform an extensive review of the literature about leishmaniasis relapses in children. Atypical mucosal involvement during Leishmaniasis relapse is an eventuality in pediatric patients. Clinical follow-up and periodic PCR tests must be considered essential for the early recognition and treatment of an eventual relapse.

## 1. Introduction

Coronavirus disease 2019 (COVID-19) is a pandemic infection whose outbreak in 2020 has complicated the management of other infectious diseases worldwide [1].

In children, COVID-19 is usually mild, and a low number of severe cases has been described when compared with adults [2]

Pediatric clinical features are usually nonspecific and include fever, cough or gastrointestinal symptoms. Laboratory findings can report a transitory lymphopenia [3].

Coinfections represent a challenge, as they can lead to an inappropriate diagnosis and management of other life-threatening conditions, thus delaying their treatment. 

Leishmaniasis is a vector-borne disease caused by obligate intracellular protozoa of the genus Leishmania [4].

Leishmania parasites are responsible for a broad variety of clinical manifestations, ranging from cutaneous and often self-healing lesions to mucocutaneous and life-threatening visceral diseases [5].

Visceral leishmaniasis (VL) is classically caused by *L. donovani* in Asia and Africa and by *L. infantum* in the Mediterranean region, where it is transmitted to humans through the bite of infected female Phlebotomine sandflies. 

In Sicily, the largest island in the Mediterranean Sea, VL is endemic [6].

If left untreated, the disease can eventually lead to death, while adequate treatment usually leads to full recovery. Even if antimonial compounds have been considered the gold standard treatment of leishmaniasis for decades [7,8], lipid formulations of amphotericin B, and particularly liposomal amphotericin B, are considered today as the treatment of choice for VL in many regions [9,10], and data have proved this efficacy and safety, even for cutaneous and mucocutaneous leishmaniasis [11].

Recurrency, rather than reinfection, is usually the most frequent mechanism of relapse [12,13,14], which is extremely rare in immunocompetent individuals and usually occurs within 6 months of treatment.

In this paper, we describe an unusual case of VL relapse with gingival localization in a child who was recently infected with SARS-CoV-2. Starting from our case, a review of the literature was carried out with the aim of analyzing the clinical characteristics, therapeutic strategies and outcomes of relapsing VL caused by *L. infantum* with mucosal involvement in pediatric patients. 

## 2. Materials and Methods

A computerized literature review was performed by submitting the following keywords into the PubMed search engines: visceral AND (Leishman* OR *L. infantum* OR Leishmania infantum) AND (oral OR mucosal OR gingiv* OR laryngeal) AND (recurrent OR relapsed OR recidiva*) AND (pediatric OR paediatric OR child* OR infant OR baby).

No language filters or other restrictions were applied to the results. Furthermore, additional relevant papers were found through citation mining.

An article was considered eligible for our review when describing cases of relapsing VL caused by *L. infantum* with mucosal localization in patients aged 0–18 years. Papers without complete clinical data or the full text available were excluded. 

For each case, the following epidemiologic and clinical variables were assessed: sex, age, clinical manifestation, underlying conditions, biological sample in which *L. infantum* was isolated at the time of the recurrence, and the therapy regimen.

## 3. Results

A total of 21 papers, dating from 1998 to 2021, were found and screened as a result of the literature review.

The assessment for eligibility led to the exclusion of 18 papers because they were not inherent to the topic of our research. An additional paper was excluded because complete data were not accessible.

In conclusion, only two articles were selected for our paper. The flow diagram in Figure 1 illustrates this selection process.

Jeziorsky et al. [15] described in 2009 the case of a 7-years-old girl treated with antitumor necrosis factor-alpha (anti-TNFα) for juvenile idiopathic arthritis, presenting with visceral leishmaniasis caused by *L. infantum* approximately two years after starting the treatment. An examination of bone marrow aspirates confirmed VL. Six doses of intravenous liposomal amphotericin B, for a total of 24 mg/kg, were administered to the patient. The outcome was favorable, and the anti-TNFα treatment was maintained. Despite prophylactic treatment with amphotericin B at 3 mg/kg/week being administered over 4 months, the child presented in 2007—26 months after the therapy with amphotericin B—with an intra-nasal granuloma with macrophages containing Leishmania parasites. The diagnosis of recurrent leishmaniasis with mucosal localization caused by *L. infantum* was confirmed by a positive result with a PCR performed on the mucosal lesion. The same authors reported in 2015 a similar case of leishmaniasis recurrence with mucosal localization after VL caused by *L. infantum* in a 4-year-old girl affected by juvenile idiopathic arthritis and bilateral uveitis treated with anti-TNFα [16].

## 4. Case Report

A previously healthy 4-year-old girl was first admitted to the Pediatric Infectious Disease Department of the Children’s Hospital of Palermo in February 2022 with hyporexia, high fever, acute onset of chest pain and a positive swab test for SARS-CoV-2. Physical examination revealed mild hepatomegaly (2 cm below the costal margin) and severe splenomegaly (5 cm below the costal margin), subsequently confirmed by an ultrasound of the abdomen, which showed a severely enlarged spleen (14.4 cm × 6.1 cm). Blood tests revealed trilinear pancytopenia with severe neutropenia (0.8 × 103/mm^3^) and severe anemia (Hb 6.6 g/dL, RBC 3.49 × 103/mm^3^, Hct 19.7%). Molecular and serological assays for Leishmania were found positive, and therapy with amphotericin B was started, with the schedule of 3 mg/kg/day for 5 days, followed by an end dose on the 10th day.

The patient was discharged after a negative Leishmania-PCR test in good clinical conditions (PCR tests were performed on the 10th and the 30th day after the beginning of the treatment). 

In September 2022, 7 months after the amphotericin B therapy had ceased, she was admitted again to the Pediatric Emergency Department of our hospital because of a one-day fever (peak 38°), hyporexia and gingival hyperemia with spontaneous bleeding, in the absence of trauma history or any evident lesion, persisting over a week. The patient had been treated at home with acyclovir without any benefit. 

A swab test for SARS-CoV-2 was performed in a hospital with a negative result returned. She appeared to be in discreet clinical condition, slightly in pain, pale, with hyperemic pharynx without plaque or purulent secretion and hypertrophic bleeding gingiva. Mild latero-cervical lymphadenopathy, mild hepatomegaly (1 cm below the costal margin) and severe splenomegaly (3 cm below the costal margin) were observed. No signs of cardiac or respiratory involvement were identified.

Because of the evidence of hepatosplenomegaly, an ultrasound of the abdomen was performed and resulted in a severely enlarged spleen (15.6 cm × 6.6 cm), a mild ascitic flap and enlarged liver. Blood tests and a complete blood count revealed trilinear pancytopenia with severe neutropenia (0.6 × 103/mm^3^), lymphopenia (1.17 × 103/mm^3^), thrombocytopenia (115 × 103/mm^3^) and anemia (Hb 7.1 g/dL, RBC 3.48 × 103/mm^3^, Hct 21.2%, RDW-CV 23.4%, RDW-SD 51.6 fL). The laboratory findings also showed hyponatremia (130 mmol/dL), hypocalcemia (8.7 mmol/dL), hypoalbuminemia (3.5 g/dL), an increase of LDH (267 IU/L), C-reactive protein (1.64 mg/dL) and alkaline phosphatase levels (190 IU/L). Additional tests were performed without significant outcomes.

With the suspicion of a relapse of leishmaniasis, PCR tests for the detection of Leishmania DNA were carried out and were found positive from both the blood sample and brush of gingival lesions. Treatment with amphotericin B was started, following the schedule of 3 mg/kg/day over 10 days. During hospitalization, the primary and acquired causes of immunosuppression were ruled out. The patient’s clinical conditions gradually improved, and the molecular diagnostic test performed on the blood sample turned negative, leading to the discharging of the patient to an outpatient follow-up.

Table 1 provides a comparison between our case and the other two cases retrieved through the literature review.

## 5. Discussion

Leishmaniasis is an important vector-borne disease that represents a serious public health problem, especially in the Mediterranean area, where it is endemic. Leishmania parasites can be responsible for a broad range of clinical manifestations, generally divided between visceral, cutaneous, and mucocutaneous leishmaniasis (MCL). 

Each clinical manifestation is typically, but not uniquely, associated with specific parasite species and depends on the immune response of the patient [17].

The majority of mucosal leishmaniasis cases are caused by *L. braziliensis*, *L. amazonensis*, *L. guyanensis* and *L. panamensis* and occur in the South American region, with a greater incidence in Peru, Bolivia, Paraguay, Ecuador, Colombia and Venezuela [18].

Sicily, as part of the Mediterranean basin, is considered to be an endemic area for VL caused by *L. infantum* [2], which is characterized by the symptomatologic triad of fever, pancytopenia, and hepatosplenomegaly. In this region, *L. infantum* can also be responsible for a localized CL, while mucosal involvement is described in the literature almost exclusively in HIV+ or immunosuppressed patients [19].

In the absence of treatment, VL is typically fatal within 2 years as a result of secondary bacterial infections or severe anemia; however, in immunocompetent patients, the treatment of the disease with amphotericin B guarantees a recovery rate that reaches almost 100% [20].

A reduction in the therapeutic response and high recurrence rates are described almost exclusively in immunocompromised patients [17,21], where Leishmania parasites can persist for decades after treatment and reactivation of the infection can occur due to an impaired cell-mediated immune response [19,22,23]. The host immune system, and in particular, the specific cell-mediated immune response, are of crucial importance in determining the clinical outcome of infection.

Several studies and experimental models suggest that the T helper-1 (Th1) response with IFN-g production induces a potent leishmanicidal mechanism in phagocytes, thus facilitating the resistance or resolution of the infection. Differently, the T helper-2 (Th2) response with IL-4 and IL-10 production can result in the inhibition of macrophage activation, with a consequent increase in the intracellular replication of the parasite and, therefore, an increased susceptibility to infection and the development of severe disease [17].

Cases of recurrence with mucosal localization in patients previously treated for VL have been described mostly in adult populations with an impaired immune system (HIV+, organ transplant, immunosuppressive therapy) [19,22,24,25].

In children, sporadic cases of VL relapse have recently been described, even in immunocompetent patients [14]; however, only two pediatric cases of recurrence with localization in the nasal mucosa after VL have been reported in the literature, and both had rheumatological disease and anti-TNFα treatment [15,16].

Our patient represents, therefore, the first reported case of recurrence of VL with mucosal involvement in an immunocompetent pediatric patient without comorbidities.

The SARS-CoV-2 infection detected during the first hospitalization represents the only risk factor specific to our case, which could have impacted the immunocompetency of the patient, thus leading to a relapse with mucosal involvement.

The most described cases of coinfections with Leishmania sp. in the literature involve Plasmodium malariae and HIV. Such a trend is probably due to the insistence of these microorganisms in the same endemic regions [26].

While the effect of a cd4 decrease in patients affected by HIV is a widely recognized cause of more serious and atypical manifestations of leishmaniosis [21,24,27], in cases of coinfections with malaria, reduced levels of *P. falciparum* parasitemia have been registered when comparing to malaria-only infections, thus suggesting a protective effect of Plasmodium coinfections on the progress of *Leishmania* sp. The increased levels of pro-inflammatory cytokines (TNF-a and IFN-g) observed in the coinfected groups are probably the cause of this effect [28].

Regarding our case, studies have found that a few months after the SARS-CoV-2 infection, changes in the immune system can occur [29]. Milena Wiech et al., in a study on the remodeling of the immune system after SARS-CoV-2 infection, observed a functional remodeling of T cells occurring during recovery from severe COVID-19. More specifically, a polarization towards an exhausted/senescent state of CD4+ and CD8+ T cells is described 3 months after the infection, and a perturbation in CD4+ Treg subsets is visible up to 6 months after COVID-19 [30]. Furthermore, several types of research on the interactions between coronaviruses and parasitosis carried out during the early stages of the pandemic confirmed the presence of a cross-talk between the two types of infection. The functional remodeling of T cells, necessary to deal with the virus, may allow the parasite to escape immune surveillance, therefore leading to the reactivation of pre-existing leishmaniasis [31].

In our case, once we ruled out other possible causes of gingival bleeding, such as periodontal and hematological diseases, the suspicion of a leishmaniasis relapse led to the correct diagnosis by performing a PCR test.

Traditionally, the gold standard for diagnosis has been a microscopic examination (after Giemsa staining) or a culture on invasive samples (spleen, bone marrow or lymph node aspirates, or liver biopsy) [32]. The performances of serological diagnostic tests, despite improvements with recombinant proteins, critically depend on the clinical form of leishmaniasis and the endemic area [33].

The PCR test on peripheral blood represents today the most sensitive and less invasive diagnostic method for leishmaniasis. After the clinical recovery of a patient, the test still remains the best diagnostic tool for detecting an infection relapse before any clinical manifestation can occur. Qualitative and semiquantitative PCRs should thus be adopted as the standard method for monitoring the response to treatment in immunocompetent children with a periodicity of 3, 6, 9 and 12 months [27,34].

Liposomal amphotericin B (AmBisome) is the only drug approved by the U.S. Food and Drug Administration for the treatment of VL in immunocompetent patients, according to a dosing schedule of seven administrations on days 1–5, 14 and 21 at a dose of 3 mg/kg/day [9]. Numerous studies have demonstrated the possibility of using it in immunocompetent children at a dose of 3 mg/kg/day for 6 days (one dose per day for the first 5 days and a last dose on the 10th day) [35,36]. In immunocompetent individuals, an efficient treatment reduces Leishmania amastigotes to an undetectable level. A life-long cellular immunity normally develops, suppressing residual parasites [37]. While the standard therapeutic regimen was adopted after the first diagnosis, the clinical presentation during the relapse led to the choice of proceeding with the therapy scheme for immunocompromised patients.

Recommendations about therapeutic schemes for immunocompromised patients are mainly based on evidence from European studies about the HIV-coinfected population, whose treatment failure and relapse rates are particularly high. A longer period or higher dosage of treatment is needed for this group according to the WHO and international guidelines [38].

In our case, a dose of 3 mg/kg/day of AmBisome, administered for 10 days, led to negative results of molecular tests and to the discharge of the patient. No relapse or other clinical issue has been reported during the follow-up tests.

Leishmaniasis relapse with atypical mucosal involvement represents an eventuality for pediatric patients. Clinical follow-up and periodic PCR tests performed within a year of treatment must be considered essential for the early recognition and treatment of this insidious condition.

## Figures and Tables

**Figure 1 pathogens-12-01127-f001:**
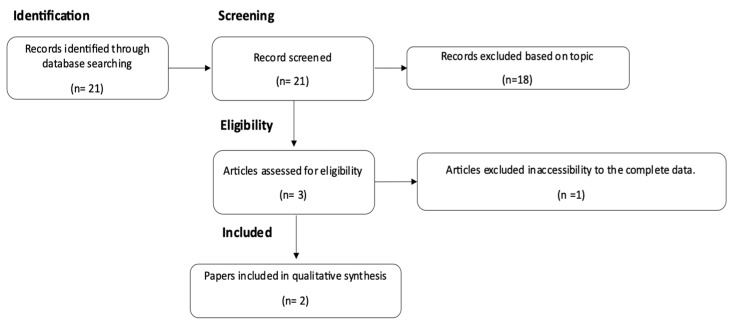
Literature review process.

**Table 1 pathogens-12-01127-t001:** Table of comparisons between our case and cases identified in the literature.

Author/Country/Year [Ref.]	Age/Sex	Pre-Existing Medical Condition	Risk Factors	VL Treatment	Time of Relapse	Location of Recurrence	Symptoms	Diagnostic Test
Jeziorski et al./France/2009 [15]	7 y/F	JIA	Immunosuppressive therapy (anti-TNFα)	L-AmB 24 mg/kg in 6 doses	Two years after treatment of VL	Nasal mucosa	Splenomegaly, recurring nosebleeds	Pathological examination and PCR on mucosal lesion
Jeziorski et al./France/2015 [16]	4 y/F	JIA, bilateral uveitis	Immunosuppressive therapy (anti-TNFα)	L-AmB 24 mg/kg in 6 doses	Two years after treatment of VL	Nasal mucosa	Splenomegaly, recurring nosebleeds	Pathological examination and PCR on the mucosal lesion
Our case Colomba et al./Italy/2023	4 y/F	COVID-19	No	L-AmB 3 mg/kg/day for 5 days, other dose on 10th day	Seven months after treatment of VL	Oral mucosa	Fever, hepatosplenomegaly, gingival bleeding	PCR on the mucosal lesion

## Data Availability

Data are contained within the article.

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
