# Peer review of "Mucosal Relapse of Visceral Leishmaniasis in a Child with SARS-CoV-2 Infection"

_pathogens, 2023, doi:10.3390/pathogens12091127_

Round 1
Reviewer 1 Report
Study present a rare case of mucosal relapse of visceral leishmaniasis in a child with Sars-Cov2 infection in children. Clinical follow up and periodic PCR tests must be considered essential for early recognition and treatment of an eventual relapse. article is important however following points must be considered before acceptence-
1- Article must discuss the clinical relevence of other co-infections in detail with citations.
2- What is difference between pathogen diversity (becteria and virus etc) discussed in terms of imnnue-suceptibility and candidate markers.
3- Other leishmania specifies in different geographical locations must be cited with pathogenic perspective. This will help to understand pathogenic risk.
4- What is public health relevence of this study. How can this study is useful for making control strategese.
5- Graphics give better idea of study in simplest form.
Category- Minor revision
Reviewer 2 Report
In this manuscript, the authors present a clinical case of visceral leishmaniasis associated with mucosal lesions in the context of recurrence. The data are interesting, but the manuscript is not sufficiently written to clarify several points. Below are my recommendations and questions.
1. The title of the manuscript refers to VL mucosal failure in a situation of co-infection with SARS-CoV-2, however in the introduction there is no mention of this viral infection, which may place the reader in spite of a possible immunosuppression or immunoactivation in SARS infection -CoV-2. I suggest rewrite the introduction, including a paragraph regard SARS-CoV-32 infection.
2. Regarding the clinical case itself, did the authors carry out any investigation to rule out immunosuppression of primary origin?
3. In presenting the results, the authors refer to table 2, but I did not see any description or table 1 available in the manuscript. Please review this information.
4. Regarding the data presented in the table, in the description of the clinical case presented, the authors did not include that the patient had hepatosplenomegaly, as described on page 3, line 108. Please add this information.
5. Page 3, line 105. The authors mentioned "The patient was discharged after a negative Leishmania-PCR test in good clinical conditions". My question: How many time after the treatment, the was performed?
6. There is no discussion about the possible interference of COVID in the development of mucosal disease by Leishmania infantum. This is an important point that can make the manuscript more robust.
7. Page 4, lines 149-150 "The host immune system and in particular the specific cell-mediated immune response is of crucial importance in determining the clinical outcome of infection". The sentence is completely out of context
8. There is no reference on the pathogenesis of mucosal leishmaniasis. I suggest reviewing the bibliography and inserting references related to the topic.
Reviewer 3 Report
This interesting clinical case of a patient who developed leishmaniasis and was discharged after receiving treatment with Amphotericin B but returned with the infection again after a specific time.
I consider that this clinical case should be published after answering certain questions and clarifications:
Could you talk about the infective species in Latin America?
in line 36 of the introduction I consider that the first line treatment is not AmB but antimmonium salts, AmB is second line treatment. Please correct this.
What about other treatments in Italy, they only conceive the idea of giving Amphotericin B
Line 42, is SARS-CoV-2 in capital letters, correct throughout the manuscript
Could you provide ultrasound and PCR images indicating that the molecular diagnosis was for Leishmania?
Could you post images of leishmaniasis in the mucosa?
They have the consent signed by the patient's parents so that the case can be published
Do you think there is any correlation with SARS-Cov-2 and leishmaniasis infection?
Minor editing of the English language required
Reviewer 4 Report
The authors are submitting a manuscript for publication titled “Mucosal Relapse of Visceral Leishmaniasis in a child with Sars-CoV2 Infection”. As far as I know, this case report is the first description of such “interaction”. The paper is well written, and I would recommend its publication after a few minor corrections are made:
· Line 211-212: the authors mention “the follow-up tests”. It would be interesting for the scientific and medical community to know for how long those follow-up tests have been carried out.
· All species names must be italicised, the authors seem to have forgotten that after page 1 and that also includes SARS-CoV2.
· Line 14: correct “genius” for “genus”
· Line 195: correct “LV” for “VL”
Round 2
Reviewer 2 Report
The authors accepted the sugestions and rewritten some parts of the manuscript. I consider the manuscript accept to be published.